# The Protective Role of Glutathione against Doxorubicin-Induced Cardiotoxicity in Human Cardiac Progenitor Cells

**DOI:** 10.3390/ijms241512070

**Published:** 2023-07-28

**Authors:** Eun Ji Lee, Woong Bi Jang, Jaewoo Choi, Hye Ji Lim, Sangmi Park, Vinoth Kumar Rethineswaran, Jong Seong Ha, Jisoo Yun, Young Joon Hong, Young Jin Choi, Sang-Mo Kwon

**Affiliations:** 1Laboratory for Vascular Medicine and Stem Cell Biology, Department of Physiology, Medical Research Institute, School of Medicine, Pusan National University, Yangsan 50612, Republic of Korea; easy2697@naver.com (E.J.L.); jangwoongbi@naver.com (W.B.J.); wozh1304@naver.com (J.C.); dla9612@naver.com (H.J.L.); smpark7257@gmail.com (S.P.); vinrebha@gmail.com (V.K.R.); hajongseong@gmail.com (J.S.H.); jsyun14@hanmail.net (J.Y.); 2Convergence Stem Cell Research Center, Pusan National University, Yangsan 50612, Republic of Korea; 3Department of Cardiology, Chonnam National University School of Medicine, Chonnam National University Hospital, Gwangju 61469, Republic of Korea; hyj200@hanmail.net; 4Department of Hemato-Oncology, School of Medicine, Pusan National University, Yangsan 50612, Republic of Korea

**Keywords:** doxorubicin, glutathione, human cardiac progenitor cells, cardiotoxicity

## Abstract

This study investigated the protective effect of glutathione (GSH), an antioxidant drug, against doxorubicin (DOX)-induced cardiotoxicity. Human cardiac progenitor cells (hCPCs) treated with DOX (250 to 500 nM) showed increased viability and reduced ROS generation and apoptosis with GSH treatment (0.1 to 1 mM) for 24 h. In contrast to the 500 nM DOX group, pERK levels were restored in the group co-treated with GSH and suppression of ERK signaling improved hCPCs’ survival. Similarly to the previous results, the reduced potency of hCPCs in the 100 nM DOX group, which did not affect cell viability, was ameliorated by co-treatment with GSH (0.1 to 1 mM). Furthermore, GSH was protected against DOX-induced cardiotoxicity in the in vivo model (DOX 20 mg/kg, GSH 100 mg/kg). These results suggest that GSH is a potential therapeutic strategy for DOX-induced cardiotoxicity, which performs its function via ROS reduction and pERK signal regulation.

## 1. Introduction

Doxorubicin (DOX) is an anticancer agent belonging to the anthracycline class of drugs that is used to treat breast cancer, bladder cancer, Kaposi’s sarcoma, lymphoma, and acute lymphocytic leukemia [1,2]. However, its application is associated with side effects, such as myocardial dysfunction, dilated cardiomyopathy, and heart failure [3,4]. The mechanism of DOX-mediated side effects is multifactorial, including intercalation of DNA, reactive oxygen species (ROS) generation, and mitochondrial dysfunction [5,6]. Among these, we focused on ROS reduction, which has been the most studied.

The adult heart is well known as an organ where differentiation is completed. Since the beginning of the 2000s, research on stem cells present in the heart has been intensively studied [7,8,9,10]. Cardiac progenitor cells (CPCs) were discovered in adult mouse hearts by Anversa et al. in 2003 [11]. The human CPCs are self-renewing, clonogenic, and multipotent as that they can differentiate into cardiomyocytes, smooth muscle cells, and endothelial cells [12]. In addition, hCPCs can proliferate and differentiate into damaged cardiomyocytes and contribute to regeneration [11,13,14]. Therefore, cardiomyocyte recovery is vital; it is achieved by protecting or restoring hCPCs in the damaged heart.

Glutathione (GSH) is an antioxidant comprising amino acids including glutamic acid, cysteine, and glycine and its reduced form protects cells by reducing the levels of ROS. Previous studies have shown that GSH protects against DOX-induced myocardial toxicity [15,16] and a recent study has investigated the protective effect of GSH against DOX-induced toxicity using animal experiments [17]. However, information regarding GSH and DOX is limited and the underlying molecular mechanisms are unknown.

Previous studies have reported that DOX-induced cardiotoxicity leads to the generation of ROS, including peroxide, superoxide, and hydroxyl radicals, and the blocking of calcium channels [18,19]. DOX-damaged cardiomyocytes, such as AC16 and H9C2, have been reported to be associated with diverse molecular mechanisms involving the Nrf2, AKT, and ERK pathways [20,21,22,23]. In addition, research on cancer cells and DOX has progressed steadily [24,25]. The alleviation of DOX-induced cardiomyopathies using hCPCs has also been studied [26]. However, the mechanism underlying the treatment with antioxidants in hCPCs damaged by DOX remains unclear.

In this study, we hypothesized that GSH, an antioxidant, relieves DOX-induced damage caused by DOX in hCPCs. We also suggest that these findings will help us to understand the mechanism by which DOX-induced damage is restored by GSH treatment.

## 2. Results

### 2.1. Effect of DOX and GSH in hCPCs Viability

To confirm whether hCPCs were damaged by DOX, they were exposed to 100, 200, 500, 1000, 2000, 4000, and 5000 nM DOX for 24 h. Cell viability was analyzed using the CCK-8 assay kit (Figure 1A). The cell viability in the DOX group was lower than that in the control group. DOX of 500 nM was the preferred concentration based on morphological changes and cytotoxicity evaluation. The hCPCs were exposed to GSH concentrations of 0.1 and 1 mM, resulting in an increase in cell viability with concentrations (Figure 1B). To confirm the recovery effect of GSH, we determined the cell viability of hCPCs exposed to 500 nM DOX and GSH at 0.1 and 1 mM concentrations. GSH had a protective effect on damaged hCPCs (Figure 1C). Numerous studies have explored the effects of different conditions by conducting experiments in serum-free environments to understand their potential impact on cell behavior. In line with this, we performed cell viability assessments in a serum deprivation setting to verify the significance of serum conditions. Remarkably, our findings exhibited a similar pattern of results in both settings, suggesting that serum components did not significantly influence the drug’s efficacy (Appendix A). It is important to note that attempting to induce serum deprivation in cardiac progenitor cells led to an exceptionally low survival rate, making it challenging to carry out further experiments including functional evaluations. In order to address these limitations and ensure the feasibility of our investigations, we extensively reviewed relevant literature [26,27]. The majority of these references had conducted their experiments in environments containing serum. Considering the well-established difficulties with serum deprivation and the wealth of supportive evidence from the literature, we proceeded with conducting all experiments in environments containing serum. It also improved the cell confluency (Figure 1D) and it did not lead to morphological differences (Figure 1E). In order to confirm cell proliferation, as a result of counting the number of cells it was confirmed that the number of cells decreased during DOX treatment and was significantly recovered through GSH treatment. In addition, as a result of Western blotting it was confirmed that increased Cyclin E and decreased CDK4 caused by DOX treatment was recovered when treated with GSH (Figure 1F,G and Appendix A). During the process of establishing the optimal concentration of GSH treatment, the treatment with GSH led to an increase in cell viability (Figure 1B). To ascertain whether the enhanced viability in the presence of DOX-GSH is solely due to the protective effect of GSH or a standalone effect, an experiment was conducted following treatment with GSH alone. The results revealed that GSH treatment did not affect significant alterations in cell confluency or cell number (Appendix A) and there were no changes observed in cell cycle-related markers or major signal transduction (Appendix A). These findings indicate that the addition of GSH in a normal cellular environment does not enhance the cell proliferative capacity or functionality significantly. However, it requires co-processing of GSH in a specific environment such as DOX.

### 2.2. Effect of GSH in Apoptosis of hCPCs by DOX

To determine whether cell death was caused by DOX-induced apoptosis, we evaluated hCPCs’ death using annexin V/PI staining (Figure 2A). In this experiment, the initial DOX concentration of 500 nM, as determined in Figure 1, resulted in lower cell viability, making it difficult to conduct further experiments. Experiments were conducted with 250 nM to confirm the apoptosis and antioxidant of cells caused by DOX. DOX treatment significantly increased the number of apoptotic cells and reduced the number of live cells. In contrast, the DOX with GSH-treated groups attenuated DOX-induced hCPCs apoptosis (Figure 2B,C). We also performed an experiment comparing the GSH-alone treated group to the control (Appendix A). As a result, no significant difference between the two groups could be confirmed. These data suggest that the reduced cell viability in Figure 1 was caused by apoptosis.

### 2.3. Effect of GSH on Generated ROS Caused DOX in hCPCs

ROS generation is well known to occur when cells are exposed to DOX. Therefore, we investigated whether DOX-induced apoptosis in hCPCs is related to ROS generation. Although ROS increased in the DOX-treated group, co-treatment of hCPCs with DOX and GSH significantly decreased cellular ROS levels (Figure 2D). A graph quantifying H_2_DCFDA is shown below (Figure 2E). These data suggest that apoptosis caused by DOX in hCPCs is induced by ROS, which is restored by GSH.

### 2.4. Recovery Effect of GSH Related to pERK in hCPCs

Based on the results shown in Figure 2, DOX-induced cytotoxicity was induced by ROS. Therefore, we determined the mechanism by which GSH could be recovered. These results suggested that while the amount of pERK in the cells increased when exposed to DOX, it decreased in the hCPCs after co-treatment with DOX and GSH, as reported by previous studies [28,29]. Upon exposing the cells to the U0126, an ERK inhibitor, an increase in the survival rate of the cells was observed which declined in viability owing to DOX (Figure 3A,B and Appendix A). Similarly to the previous findings, the cell group treated with GSH did not exhibit a significant difference compared to the control group (Appendix A). These data demonstrated that GSH was restored through ERK signaling.

### 2.5. Effect of GSH on Cell Migration and Tube Formation Capacity Impaired by DOX in hCPCs

To evaluate the function of hCPCs after exposure to DOX, we selected a 100 nM concentration of DOX that did not affect the viability and exposed the cells treated with 100 nM DOX to GSH to conduct a wound healing assay (Figure 4A,B). The migration ability was reduced when the cells were treated with DOX alone but was restored in the group treated with DOX and GSH (Figure 4C,D). A transwell migration assay was performed to confirm these results. The group exposed to GSH showed an increase in migration ability compared with the DOX group. Tube formation assays were performed to evaluate the other functions (Figure 4E,F). Similar results were observed in the tube formation experiments. These data suggest that GSH restores cell migration and tube formation impaired by DOX in hCPCs.

### 2.6. Effect of GSH and DOX In Vivo

To confirm whether GSH can inhibit DOX-induced cardiotoxicity, DOX (20 mg/kg, i.p.) or GSH (100 mg/kg, i.p.) was administered. The survival rate and body weight were measured daily (Figure 5A). As a result, the DOX-treated group exhibited a lower survival rate and body weight compared to the control and DOX with GSH-treated groups (Figure 5B,C). After harvesting the heart of the surviving mice on day 6 and sectioning the tissue, qRT-PCR, Masson trichrome staining (M and T), and hematoxylin and eosin (H and E) staining were performed. Bnp and Myh7 markers related to cardiotoxicity were identified at the mRNA level in the mouse heart. As a result, it was possible to confirm the decreased results in the DOX + GSH group compared with the increased results in the DOX group (Figure 5D). In addition, immunohistochemistry confirmed that fibrosis occurred at the injured site in the heart tissue of mice in the DOX group; however, it was ameliorated in the DOX + GSH group (Figure 5E). These results suggested that GSH protects against DOX-induced cardiotoxicity.

## 3. Discussion

DOX, an anticancer agent, exerts its cardiotoxic effects through multiple mechanisms involving not only ROS but also calcium dysregulation and mitochondrial dysfunction. While these factors collectively contribute to cardiotoxicity, ROS can be considered a prominent factor in this process [30,31]. Consequently, numerous researchers have endeavored to prevent these side effects; however, specific strategies for prevention and alleviation are lacking. Previous studies indicate that DOX induces oxidative free radical production and reduces the expression of antioxidant enzymes in the heart and mouse myocardial cells [32,33,34]. In this study, we identified GSH that enhanced hCPCs’ viability and function against DOX-induced cardiotoxicity (Figure 1 and Figure 4). We observed cell viability and function restoration, suggesting that GSH can protect against DOX-induced cardiotoxicity.

ROS are generated during normal metabolic processes and play vital roles in cellular homeostasis, proliferation, and cell death. In our study, DOX-induced apoptosis occurred through excessive ROS production, leading to cardiotoxicity. To prevent DOX-induced ROS generation in hCPCs, we investigated naturally occurring compounds within the body rather than artificial ones. GSH is naturally synthesized and present in almost all cells in the body. It forms disulfide bonds with cysteine residues in proteins. Many researchers have studied the use of antioxidant drugs against DOX-induced side effects; however, the protective mechanism of GSH has been overlooked [15,17,26,35]. Thus, we hypothesized that GSH exerts a protective effect against DOX-induced cardiotoxicity and observed that GSH has protective effects against cardiotoxicity in vitro and in vivo [36,37]. Previous research [9,38,39] suggests that DOX treatment damages the heart. Our data emphasize the importance of modulating DOX-induced hCPCs’ dysfunction to provide a protective effect. Treating hCPCs with GSH markedly reduced DOX-induced apoptosis and cell death by enhancing CDK4 activation, as demonstrated by the results of Western blotting and Annexin V/PI staining. Additionally, GSH restored cell viability through the ERK signaling pathway and this protective effect was confirmed through in vitro results, including the migration and angiogenesis abilities of hCPCs which were reduced owing to the antioxidant effect of GSH. Furthermore, the cardioprotective effect of GSH was confirmed in mouse models in which the cardioprotective effect of GSH was demonstrated in relation to DOX-induced cardiotoxicity. Previous studies have shown that DOX-induced cardiotoxicity induces apoptosis by regulating pERK. Although we found that the protective effects of GSH in vitro are related to the pERK pathway to cardiotoxicity, the signaling mechanism by which ERK phosphorylation is involved in the detrimental effects of DOX-induced cardiotoxicity is not elucidated. However, several studies have suggested potential pathways involved in the pro-death stimulus [28,40]. ERK are known to play a role in cell survival and proliferation but their dysregulation can contribute to cell death under certain conditions. The study by Chen et al. suggested that the activation of ERK contributed to the detrimental effects of DOX on the myocardium [41]. These studies suggest that the activation of ERK may be a key signaling mechanism transducing the pro-death stimulus of DOX-induced cardiotoxicity. However, it is important to note that the exact molecular events and downstream targets of ERK in this context require further investigation for a comprehensive understanding of the signaling pathway. In summary, this study emphasizes the importance of further research to explore the potential of GSH in protecting against DOX-induced cardiotoxicity. There are several promising areas for future investigation. Firstly, exploring additional cardiac function biomarker genes and utilizing protein analysis techniques can provide deeper insights into the underlying mechanisms involved. Secondly, assessing markers of oxidative damage and mitochondrial function can help elucidate the protective effects of GSH. Additionally, non-invasive imaging techniques and pharmacokinetic studies of GSH can enhance our understanding and optimize therapeutic interventions. By pursuing these research directions, we can advance our knowledge and develop innovative strategies to mitigate the harmful effects of DOX on the heart.

## 4. Materials and Methods

### 4.1. Cell Culture

As described in a previously modified protocol, hCPCs were isolated from the human heart tissues procured after surgical procedures [7,9,42]. The Ethical Review Board of the Pusan National University Yangsan Hospital approved the protocol. The heart tissue specimens were ground into roughly 0.2 mm^3^ pieces using fine scissors in a 60-mm petri dish on ice under aseptic conditions to isolate the hCPCs. The hCPCs were digested after transferring them into 50 mL tubes containing prewarmed 0.2% collagenase type II (Worthington, NJ, USA) solution in Ham’s F-12 medium (HyClone, GE Healthcare, Chicago, IL, USA). The tubes were then incubated in a water bath at 37 °C for 1 h, with shaking every 10 min. Thereafter, single cardiac cells were filtered through a 70 µm cell strainer and centrifuged at 1200 rpm for 3 min. hCPCs were cultured in Ham’s F-12 medium containing 10% heat-inactivated fetal bovine serum (FBS; Gibco, Thermo Fisher Scientific, Carlsbad, CA, USA), 1% penicillin/streptomycin (P/S, Welgene, Daegu, Republic of Korea), 0.2 mM of L-glutathione (Sigma-Aldrich #G4251, St. Louis, CA, USA), 20 ng/mL of recombinant human basic fibroblast growth factor (rb-FGF; PeproTech, Rocky Hill, NJ, USA), and 0.005 unit/mL of human erythropoietin (hEPO; R and D Systems, Minneapolis, MN, USA). The cultures were maintained at 37 °C in a 5% CO_2_ humidified atmosphere.

### 4.2. Cell Viability Assay

The hCPCs’ viability assay was conducted using a CCK-8 kit (Dongin, CCK-3000, Seoul, Republic of Korea) according to the manufacturer’s instructions. To analyze the cell proliferation ability, 10,000 cells/well were plated into each 96-well plate along with 100–5000 nM doxorubicin hydrochloride (Sigma-Aldrich, St. Louis, CA, USA) and 0.1 and 1 mM of GSH and then incubated for 24 h after which the medium was changed to fresh drug-containing medium. After incubation, the drug-containing medium was replaced with 100 µL of CCK-8 solution. The plates were then incubated for 1 h. The absorbance was measured at 450 nm using a spectrophotometer (TECAN, Grodig, Austria). Each experiment was repeated at least thrice. The DOX concentration was selected based on what was specified in previous studies [9].

### 4.3. Apoptosis Assay

The hCPCs’ apoptosis assay was performed using an Annexin V/PI kit (BD Pharmingen, #556547, San Diego, CA, USA). The hCPCs were pre-conditioned with 250 nM DOX and 0.1 and 1 mM GSH in Ham’s F-12 medium. The concentration of 500 nM DOX was modified to 250 nM because it resulted in lower cell viability, making it difficult to conduct further experiments. After 24 h incubation, hCPCs were harvested and washed with phosphate-buffered saline (PBS), 2% FBS, and 200 µM EDTA in PBS. The pellets were suspended in 1× Annexin binding buffer with annexin V and PI according to the manufacturer’s instructions. The assay was analyzed using FACS (BD Accuri C6, BD Biosciences, San Diego, CA, USA).

### 4.4. Measurement of Intracellular ROS Levels

Intracellular ROS levels were measured using a H_2_DCFDA kit (Thermo Fisher Scientific, Carlsbad, CA, USA). The hCPCs were pre-conditioned with DOX 250 nM and GSH 0.1 and 1 mM in Ham’s F-12 medium. The cells were harvested and washed with 2% FBS and 200 µM EDTA in PBS. hCPCs were incubated and after centrifugation at 2000× *g* for 3 min, the pellet was suspended in 5 μM H_2_DCFDA in PBS containing 2% FBS and 200 μM EDTA for 10 min at 37 °C in 5% CO_2_ atmosphere. After incubation, the cells were washed with PBS and analyzed by FACS using the BD Accuri C6 software (BD Biosciences).

### 4.5. Wound Healing Assay

A six-well plate was seeded with 200,000 cells/well to investigate the migratory ability of the hCPCs. Cells were treated with 100 nM DOX, a concentration that does not induce cell death, to confirm the migration ability. Wounds were created by stroking the cells with a yellow pipette tip and the detached cells were washed with PBS. After 6 h of incubation, the migrated cells were observed at 40× magnification under a light microscope (Olympus, Tokyo, Japan). The migrated area was measured using ImageJ software (Free software from the National Institutes of Health) and was calculated using the following formula: percentage of migrated area is [(original scratched area − recovered scratched area)/original scratched area] × 100%.

### 4.6. Migration Assay

The migration assay was performed using a 24-well 8.0 μm polycarbonate transwell chamber consisting of a permeable membrane (Corning Inc., Corning, NY, USA). First, hCPCs were pretreated to 100 nM of DOX with 0.1 and 1 mM of GSH. For the assay, 500 μL of Ham’s F-12 media culture medium was added below the cell permeable membrane whereas 7000 cells/100 μL in serum-free Ham’s F-12 medium were plated on the upper chamber of the permeable membrane. After 24 h of incubation, the migrated cells were fixed with 4% paraformaldehyde and stained with 0.5% crystal violet at room temperature. The upper chamber was washed and the top of the membrane was examined for cell migration. After mounting, the cells were observed under an inverted microscope and the number of cells was counted.

### 4.7. Tube Formation Assay

A tube formation assay was performed to assess the function of the hCPCs in the formation of blood vessel-like structures (tubes). DOX and GSH concentrations were the same as those treated in the migration and wound healing assays. As for the 96-well plates, they were coated with 65 μL of Matrigel (BD Biosciences, San Diego, CA, USA) and incubated at 37 °C for 30 min. After incubation, 7000 cells were seeded into a 96-well plate coated with Matrigel, incubated for 7 h, and analyzed every hour to examine the tube-forming ability. After incubation, the total tube length was measured using ImageJ software.

### 4.8. Western Blotting

After culturing in each group media, the cells were lysed using RIPA lysis buffer (Thermo Fisher Scientific, Waltham, MA, USA) supplemented with either a protease inhibitor cocktail or protease and phosphatase inhibitor cocktail (Thermo Fisher Scientific, Waltham, MA, USA). After 30 min of reaction at 4 °C, the cells were centrifuged at 13,000× *g* for 30 min. After transferring the supernatant to a new 1.5 mL white e-tube, the protein concentrations were quantified using a Bicinchoninic Acid Kit buffer (Thermo Fisher Scientific, Waltham, MA, USA). The proteins in each group were loaded at 30 μg, separated using 8–15% sodium dodecyl sulfate polyacrylamide gel electrophoresis (SDS-PAGE), and transferred to a polyvinylidene fluoride membrane (PVDF) at 300 mA for 2 h (Millipore, Billerica, MA, USA). After the membrane was blocked in 5% skim milk for 1 h at room temperature, it was incubated overnight at 4 °C with primary antibodies specific for Cyclin D1 (1:200, Santa Cruz Biotechnology, Dallas, TX, USA; SC-8396), CDK4 (1:200, Santa Cruz, SC-56277), ERK (1:1000, Cell Signaling, 4695S), pERK (1:1000, Cell Signaling, 4376S), AKT (1:1000, Cell Signaling, 4691S), pAKT (S473) (1:1000, Cell Signaling, 4060S), and β-actin (1:5000, Santa Cruz, SC-47778). The membranes were washed thrice with Tris-buffered saline containing 0.1% Tween 20 (TBST) and were incubated with horseradish peroxidase-conjugated anti-rabbit IgG (1:5000, Enzo Life Sciences, Farmingdale, NY, USA) and HRP conjugated anti-mouse IgG (1:5000, Enzo Life Sciences, NY, USA) secondary antibodies for 1 h at room temperature. After washing the membranes again with TBST, the bands were visualized using Luminate Crescendo Western HRP Substrate (Millipore, Billerica, MA, USA) and an X-ray film. *Β*-Actin was used as the loading control for Western blotting.

### 4.9. DOX-Induced Cardiomyopathy Model

Experiments were performed on 6 to 10-week-old male C57BL/6 mice maintained under a 12 h light/dark cycle following the regulations of the Pusan National University. All experiments were performed in accordance with the Pusan National University Institutional Animal Care and Use Committee (PNUIACUC). For the DOX-induced cardiomyopathy model, C57BL/6 mice were randomized into four groups and intraperitoneally injected with PBS (*n* = 10), DOX (*n* = 10, 20 mg/kg), GSH (*n* = 10, 100 mg/kg), or DOX with GSH (*n* = 10, 100 mg/kg), as previously described [17,43,44]. Their body weight and survival rate were measured daily after the injection of DOX and GSH.

### 4.10. Histological Staining

Mice were euthanized five days after treatment and their heart tissue was retrogradely perfused with PBS and fixed with 4% paraformaldehyde overnight at 4 °C. Tissue sections (5 µm thickness) were subjected to histological staining with M and T and H and E staining. Sections were examined using a Lionheart FX automated microscope (BioTek, Winooski, VT, USA).

### 4.11. Quantitative Real-Time PCR

The total RNA was isolated from the experimental cell groups and mouse heart tissue using TRIzol reagent (Thermo Fisher Scientific, Waltham, MA, USA) according to the manufacturer’s specifications. Reverse transcription was performed on the total RNA (1 µg) using the PrimeScript™ first strand cDNA synthesis kit (Clontech, TaKaRa 6110A, Mountain View, CA, USA). PCR was performed according to the manufacturer’s protocol. Real-time PCR was performed on an Applied Biosystems 7500 Real-time PCR system (Thermo Fisher Scientific, Waltham, MA, USA) using SYBR Green PCR Master Mix (Thermo Fisher Scientific, Waltham, MA, USA). The qRT-PCR protocol was begun with 95 °C incubation for 10 min and then followed by 45 cycles of 95 °C for 10 s, 60 °C for 10 s, and 72 °C for 10 s. It was subsequently followed by melting at 95 °C for 10 s and 60 °C for 60 s. Specific primers for mMyh7 (forward primer, 5′-GCTGAAAGCAGAAAGAGATTATC-3′, and reverse primer, 3′-TGGAGTTCTTCTCTTCTGGAG-5′), mBnp (forward primer, 5′-AAGTCCTAGCCAGTCTCCAGA-3′, and reverse primer, 3′-GAGCTGTCTCTGGGCCATTTC-5′), and β-actin (forward primer, 5′-TCAGGTCATCACTATCGGCAATG-3′, and reverse primer, 3′-TTTCATGGATGCCACAGGATTC-5′) were used. All samples were analyzed for quantification using the double delta Ct method (2^–∆∆Ct^). We used a single-peak melting curve analysis to confirm the value obtained from real-time PCR.

### 4.12. Statistical Analysis

Statistical analyses were conducted using a two-tailed unpaired Student’s *t*-test and a one-way analysis of variance (ANOVA) using GraphPad Prism software (GraphPad, Inc., La Jolla, CA, USA). Data are reported as mean ± standard deviation. Differences were considered statistically significant at *p* < 0.05.

## 5. Conclusions

Our study highlights the protective effects of GSH against DOX-induced cardiotoxicity mediated through the modulation of the pERK signaling pathway. GSH restores cell viability and function, attenuates ROS generation, and mitigates apoptosis in hCPCs. These findings suggest that GSH may serve as a promising treatment strategy to alleviate the cardiotoxic side effects associated with DOX treatment. However, further research is needed to unravel the molecular mechanisms underlying the interaction between GSH and pERK signaling, paving the way for developing targeted therapeutic interventions for DOX-induced cardiotoxicity (Figure 6).

## Figures and Tables

**Figure 1 ijms-24-12070-f001:**
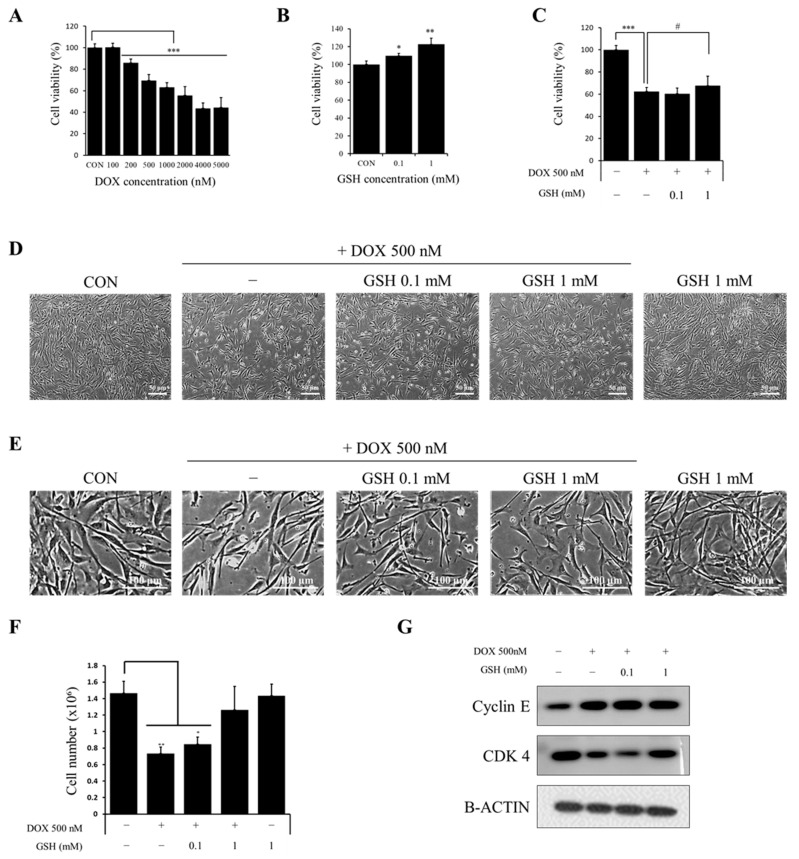
The effect of DOX and GSH in hCPCs viability. (**A**) hCPCs viability in various concentrations of DOX. (**B**) hCPCs viability in various concentrations of GSH. (**C**) hCPCs were exposed to DOX 500 nM and GSH as a combination. (**D**) Cell morphology and (**E**) confluency after exposure to DOX alone and with GSH (**F**) Cell number. (**G**) Western band detection related to cell cycle. Values are expressed as the mean ± standard derivation. * *p* < 0.05, ** *p* < 0.01, and *** *p* < 0.001 compared to the control group and # *p* < 0.05 compared to the DOX group.

**Figure 2 ijms-24-12070-f002:**
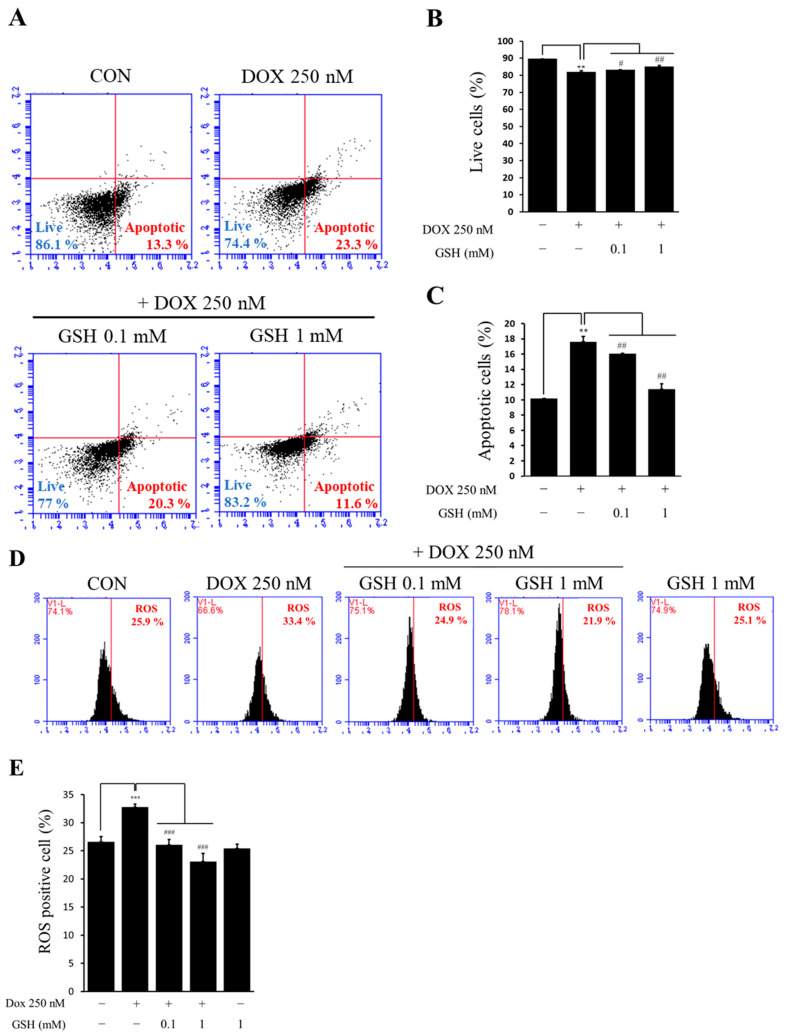
Effect of GSH on hCPCs apoptosis and ROS generation caused by DOX. (**A**) Apoptosis was measured by FACS and using an annexin V/PI staining kit. Based on FACS, data were quantified at apoptotic cells (**B**) and live cells (**C**). (**D**) ROS generation was measured by FACS and H_2_DCFDA. (**E**) It was quantified to the ROS generation rates of FACS data. Data are expressed as the mean ± standard derivation. ** *p* < 0.01, and *** *p* < 0.001 compared to the control group and # *p* < 0.05, ## *p* < 0.01, and ### *p* < 0.001 compared to the DOX group.

**Figure 3 ijms-24-12070-f003:**
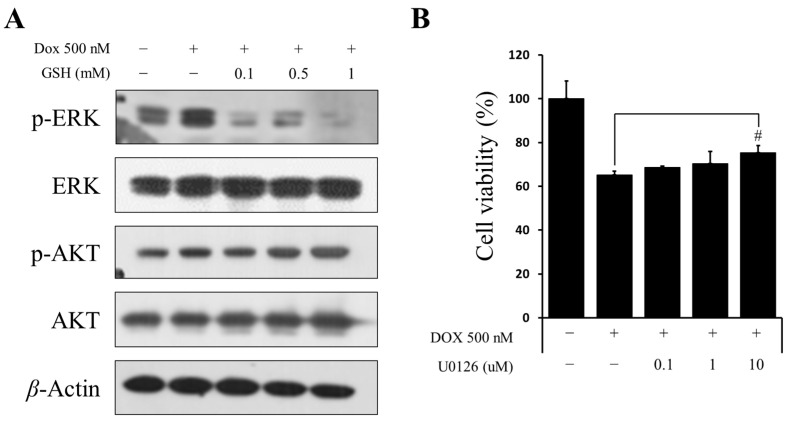
Recovery effect of GSH on pERK in hCPCs. (**A**) The expression level of proteins related to ERK and AKT signaling. (**B**) Cell viability after DOX exposure and treatment with U0126 as an ERK inhibitor. Data are expressed as the mean ± standard derivation. # *p* < 0.05 compared to the DOX group.

**Figure 4 ijms-24-12070-f004:**
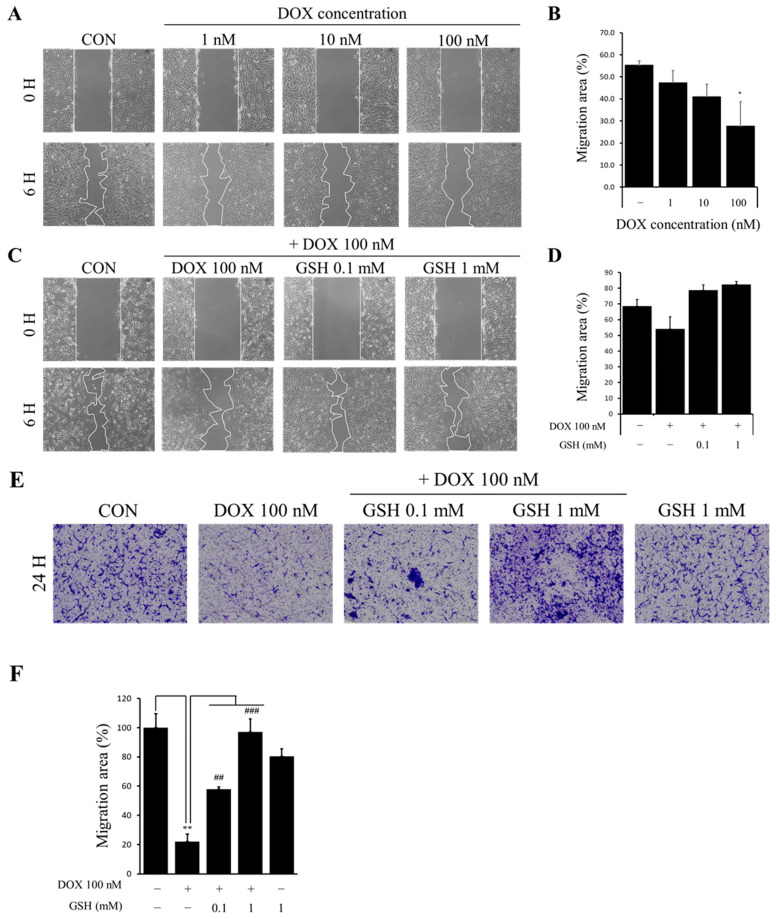
GSH enhances the migration and tube forming capacity impaired by DOX in hCPCs. (**A**) The wound healing assay for selecting the appropriate proper concentration did not affect the cell viability. (**B**) Data were quantified. (**C**) hCPCs were exposed to 100 nM DOX and GSH alone or together. The wound healing assay for the effect of GSH and DOX. (**D**) Quantification of the wound healing assay. (**E**) The migration function by using a transwell migration kit. (**F**) Quantification of a transwell migration assay. (**G**) The tube formation ability of hCPCs exposed to DOX and GSH. (**H**) Quantification of the tube length. Data are expressed as the mean ± standard derivation. * *p* < 0.05, ** *p* < 0.01, and *** *p* < 0.001 compared to the control group and # *p* < 0.05, ## *p* < 0.01, and ### *p* < 0.001 compared to the DOX group. N.S.: non-significance.

**Figure 5 ijms-24-12070-f005:**
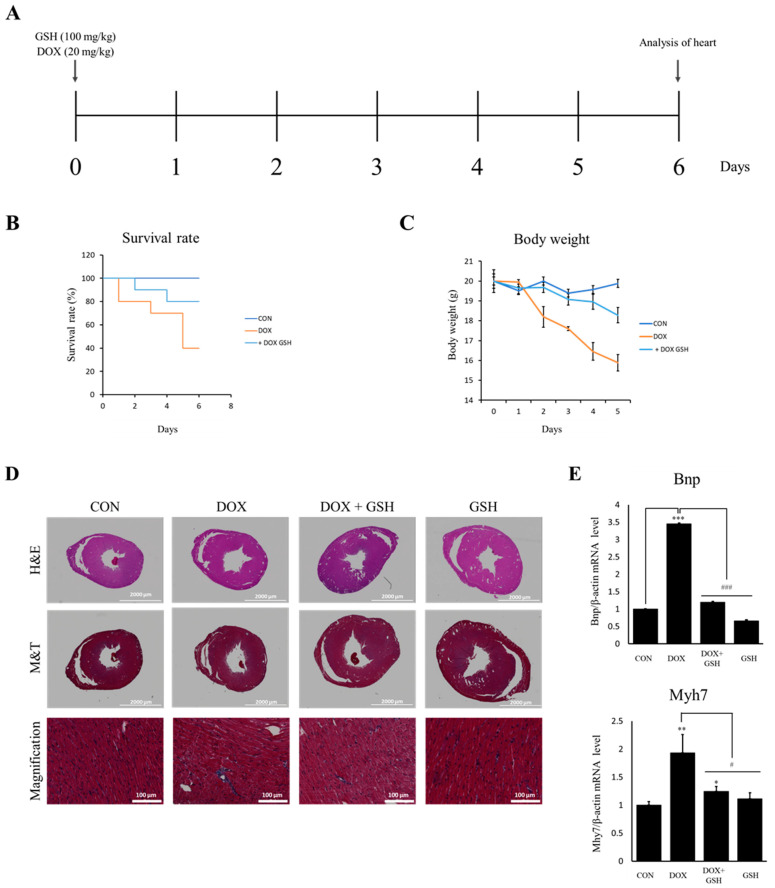
Effect of GSH and DOX in vivo. (**A**) Schematic representation of in vivo experimental design. (**B**) Representative image of H and E staining and M and T staining at 6 days after IP injections of DOX and GSH. (**C**) Representative body weight and Kaplan–Meier estimator at 5 days after IP injections of DOX and GSH. (**D**) The cardiac tissues of each group were identified through immunohistochemistry. (**E**) Bnp and Myh7, markers of mRNA expression of cardiotoxicity, were measured by qRT-PCR. Data of qRT-PCR are expressed as mean ± standard deviation (S.D). * *p* < 0.05, ** *p* < 0.01, and *** *p* < 0.001 compared to the control group and # *p* < 0.05 and ### *p* < 0.001 compared to the DOX group.

**Figure 6 ijms-24-12070-f006:**
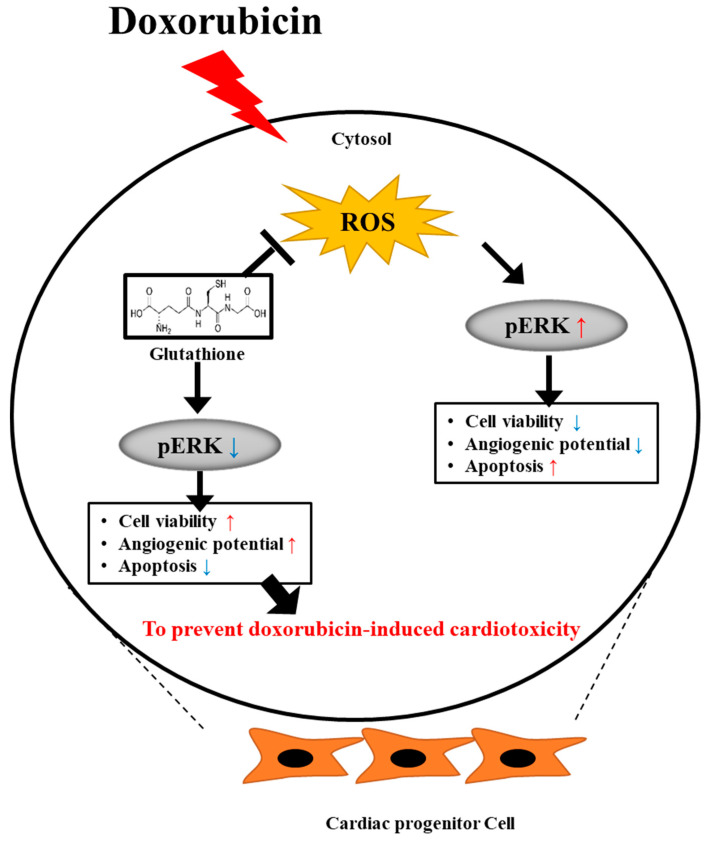
Proposed working model. GSH contributes to prevent DOX-induced cardiotoxicity though the pERK pathway. Colored-arrows: In this figure, red arrows indicate an increase, while blue arrows signify a decrease.

## Data Availability

The data used to support the findings of this study are included in the article.

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
