# Peer review of "The Protective Role of Glutathione against Doxorubicin-Induced Cardiotoxicity in Human Cardiac Progenitor Cells"

_ijms, 2023, doi:10.3390/ijms241512070_

Round 1
Reviewer 1 Report
The manuscript by Lee and co-workers addresses an interesting and pertinent topic, since it is relevant to develop and establish treatment and protective approaches to DOX-induced cardiotoxicity.
This manuscript reports a well-reasoned study, combining in vitro and in vivo approaches, as well as a number of complementary techniques, to characterize the alleviating effects of GSH on DOX-induced cardiotoxicity. The results have potential applicability in the clinical settings, which represents an added value. The draft is well structured.
Despite its current flaws, which I detail below, I believe that this paper fits the scope of the IJMS.
Title
· Please replace “progeniter” by “progenitor”.
Abstract
· This section should be complemented with information on DOX and GSH treatments (duration and concentrations used).
Introduction
· Lines 28-29: “we focused on ROS inhibition” – I believe “we focused on ROS reduction” (or an analogous expression) would be more accurate. Please rephrase this sentence.
· Lines 45-47: “Previous studies have reported that DOX-induced cardiotoxicity leads to (…) blocking of calcium channels by antioxidants” – I believe the effects on calcium channels is unclear; is it a result from DOX exposure or from the exposure to antioxidants? Please rephrase this sentence.
Results
· Line 68: what is meant by “the cell cycle was confirmed to be western”? Please rephrase this sentence.
· Figure 2E: a label must be provided for the vertical axis of the graphic.
Materials and Methods
· Please specify the DOX and GSH concentrations used in all assays. A rationale for the choice of these concentrations should also be provided.
· Section 4.8: please specify the amount(s) of protein analyzed per lane, the conditions of protein transfer to the PDVF membranes and the secondary antibody dilution(s) and brand(s).
· Section 4.9: which was the rationale for the DOX dose used to create the cardiomyopathy model, as well as the rationale behind the selection of GSH dose? Please add some information on this, as well as on the choice of the route of administration of these compounds vs. their commonest route(s) of administration.
· Section 4.11: “Immunohistochemistry” is not the right choice of word, since no use of antibodies is mentioned within this section. Maybe “histological staining” or “histological analysis” would be more accurate.
· Section 4.12, line 321: “- delta(delta Ct)” should appear as a superscript.
· Section 4.12: the genes analyzed, as well as the primer sequences and the cycling conditions used for their amplification, should be specified.
· Why weren’t serum BNP levels quantified, since it is a common way to characterize cardiac function? Please briefly elaborate on this.
Discussion
· Please acknowledge future prospects/opportunities for complementary assays. For instance, additional cardiac function biomarker genes and proteins (and the respective analyzing techniques) can be suggested.
I believe some English language editing is required.
Author Response
Title
- Comment and Suggestion for Authors: Please replace “progeniter” by “progenitor”.
Answer: We thank you for pointing this out. We apologize for the mistake in labeling the title. We have corrected this error in the revised manuscript.
Revised text line (3): The Protective Role of Glutathione Against Doxorubicin-Induced Cardiotoxicity in Human Cardiac Progenitor Cells
Abstract
- Comment and Suggestion for Authors: This section should be complemented with information on DOX and GSH treatments (duration and concentrations used).
Answer: Thank you for the opportunity to improve the explanation through your advice. We revised the abstract part by adding the on DOX and GSH treatments (duration and concentrations used) according to your suggestion.
Revised text line (13 – 22): Abstract: This study investigated the protective effect of glutathione (GSH), an antioxidant drug, against doxorubicin (DOX)-induced cardiotoxicity. Human cardiac progenitor cells (hCPCs) treated with DOX (250 to 500 nM) showed increased viability and reduced ROS generation and apoptosis with GSH treatment (0.1 to 1 mM) for 24h. In contrast to the 500 nM DOX group, pERK levels were restored in the group co-treated with GSH, and suppression of ERK signaling improved hCPC survival. Similar to the previous results, the reduced potency of hCPC in the 100 nM DOX group, which did not affect cell viability, was ameliorated by co-treatment with GSH (0.1 to 1 mM). Furthermore, GSH protected against DOX-induced cardiotoxicity in the in vivo model (DOX 20mg/kg, GSH 100mg/kg). These results suggest that GSH is a potential therapeutic strategy for DOX-induced cardiotoxicity, which performs its function via ROS reduction and pERK signal regulation.
Introduction
- Comment and Suggestion for Authors:Lines 28-29: “we focused on ROS inhibition” – I believe “we focused on ROS reduction” (or an analogous expression) would be more accurate. Please rephrase this sentence.
Answer: Thank you for pointing out the need for a more accurate description of our focus in the manuscript. We corrected the word as advised.
Revised text Line (31 - 32): Among these, we focused on ROS reduction, which has been the most studied.
- Comment and Suggestion for Authors: Lines 45-47: “Previous studies have reported that DOX-induced cardiotoxicity leads to (…) blocking of calcium channels by antioxidants” – I believe the effects on calcium channels is unclear; is it a result from DOX exposure or from the exposure to antioxidants? Please rephrase this sentence.
Answer: We are very grateful for your valuable comments, which have given us the opportunity to improve our manuscript. Necessary changes have been made in the revised manuscript.
Revised text Line (47 – 49) : Previous studies have reported that DOX-induced cardiotoxicity leads to the generation of ROS, including peroxide, superoxide, and hydroxyl radicals, and blocking of calcium channels [18,19].
Results
- Comment and Suggestion for Authors: Line 68: what is meant by “the cell cycle was confirmed to be western”? Please rephrase this sentence.
Answer: We apologize for any confusion caused. We have corrected this error in the revised manuscript.
Revised text Line (72 – 74): In addition, as a result of Western blotting, it was confirmed that increased Cyclin E and decreased CDK4 caused by DOX, treatment was recovered when treated with GSH (Figure 1F, G).
- Comment and Suggestion for Authors: Figure 2E: a label must be provided for the vertical axis of the graphic.
Answer: We thank you for pointing this out. We apologize for the mistake in labeling the figures. We have corrected this error in the revised manuscript
Revised figure 2. E :
Materials and Methods
- Comment and Suggestion for Authors: Please specify the DOX and GSH concentrations used in all assays. A rationale for the choice of these concentrations should also be provided.
Answer: Thank you for your comments. Accordingly, we revised manuscripts
Revised text Line (254 - 256 and 260 – 261): 10,000 cells/well were plated into each 96 well plate along with doxorubicin hydrochloride (Sigma-Aldrich, St. Louis, CA, USA) 100 to 5000 nM and, glutathione of 0.1 and 1 mM at concentrations and incubated for 24 h
Doxorubicin concentration was selected from previous studies.
Revised text Line (282 – 284 and 292 – 294): In order to confirm the migration ability, the experiment was performed by treating 100 nM of doxorubicin, which is a concentration that does not death the cells.
First, we performed that hCPCs were pretreated to 100 nM of doxorubicin with 0.1 and 1 mM of GSH.
Revised text Line (303 – 304): Doxorubicin and glutathione concentrations are the same as those treated in the migration and wound healing assays.
- Comment and Suggestion for Authors: Section 4.8: please specify the amount(s) of protein analyzed per lane, the conditions of protein transfer to the PDVF membranes and the secondary antibody dilution(s) and brand(s).
Answer: Thank you for your feedback. As your opinion, we modified Section 4.8.
Revised text line (316 – 329 and 325 – 328): Proteins in each group were loaded at 30 μg, separated by 8-15% SDS-PAGE (sodium do-decyl sulfate polyacrylamide gel electrophoresis), and transferred to polyvinylidene fluo-ride membrane (PVDF) at 300 mA for 2 h (Millipore, Bilerica, MA, USA).
Subsequently, the membranes were washed thrice with Tris-buffered saline containing 0.1 % Tween 20 (TBST) and were incubated with horseradish peroxidase-conjugated anti-rabbit IgG (1:5000, Enzo Life Sciences, NY, USA) and HRP conjugated anti-mouse IgG (1:5000, Enzo Life Sciences, NY, USA) secondary antibodies for 1 h at room temperature.
- Comment and Suggestion for Authors: section 4.9: which was the rationale for the DOX dose used to create the cardiomyopathy model, as well as the rationale behind the selection of GSH dose? Please add some information on this, as well as on the choice of the route of administration of these compounds vs. their commonest route(s) of administration.
Answer: Thanks a lot for your comment. According to your suggestion, we added In vivo information.
Revised text Line (334 – 338) : For the doxorubicin-induced cardiomyopathy model, C57BL/6 mice were intraperitoneally injected with doxorubicin (20 mg/kg) or glutathione (100 mg/kg) based on reported studies [17,41,42].
- Comment and Suggestion for Authors: Section 4.11: “Immunohistochemistry” is not the right choice of word, since no use of antibodies is mentioned within this section. Maybe “histological staining” or “histological analysis” would be more accurate.
Answer: Thank you for your comment. We have replaced the words according to your opinion.
Revised text Line (353) : 4.11. Histological staining
- Comment and Suggestion for Authors: Section 4.12, line 321: “- delta(delta Ct)” should appear as a superscript.
Answer : Thank you for your valuable comment. We have modified manuscripts
Revised text Line (371) : For quantification, all samples were analyzed using the double delta Ct method (2– delta(delta Ct)).
- Comment and Suggestion for Authors: Section 4.12: the genes analyzed, as well as the primer sequences and the cycling conditions used for their amplification, should be specified.
Answer : We are very grateful that your valuable comments gave us the opportunity to improve our manuscript. As suggested, we have added in the manuscript
Revised text Line (366 – 371) : Specific primers for mMyh7 (forward primer, 5’- GCTGAAAGCAGAAAGAGATTATC-3’, reverse primer, 3’- TGGAGTTCTTCTCTTCTGGAG-5’), mBnp (forward primer, 5’-AAGTCCTAGCCAGTCTCCAGA-3’, reverse primer, 3’-GAGCTGTCTCTGGGCCATTTC-5’) and β-actin (forward primer, 5’-TCAGGTCATCACTATCGGCAATG-3’, reverse primer, 3’-TTTCATGGATGCCACAGGATTC-5’) were used.
- Comment and Suggestion for Authors: Why weren’t serum BNP levels quantified, since it is a common way to characterize cardiac function? Please briefly elaborate on this.
Answer: First of all, thanks for your valuable advice. As you said, we tried to measure BNP levels in serum, but we obtained a small amount of serum and were unable to measure BNP levels. For this reason, it has been replaced by mRNA level measurement in mouse heart models.
Discussion
- Comment and Suggestion for Authors: Please acknowledge future prospects/opportunities for complementary assays. For instance, additional cardiac function biomarker genes and proteins (and the respective analyzing techniques) can be suggested.
Answer: We appreciate the reviewer's suggestion regarding future prospects and opportunities for complementary assays to further enhance the understanding of the investigated topic. We approved with your suggestion and added the content of the discussion below.
Revised text Line (202 – 232) : Treating hCPCs with GSH markedly reduced DOX-induced apoptosis and cell death by enhancing CDK4 and Cyclin D1 activation, as demonstrated by the results of western blotting and Annexin V/PI staining. Additionally, GSH restored cell viability through the ERK signaling pathway, and this protective effect was confirmed through in vitro results, including the migration and angiogenesis abilities of hCPCs, which were reduced owing to the antioxidant effect of GSH. Furthermore, the cardioprotective effect of GSH was con-firmed in mouse models, in which the cardioprotective effect of GSH was demonstrated in relation to the DOX-induced cardiotoxicity. Previous studies have shown that DOX-induced cardiotoxicity induces apoptosis by regulating phosphorylated ERK (pERK). Although we found that the protective effects of GSH in vitro are related to the pERK pathway to cardiotoxicity, the signaling mechanism by which ERKs phosphorylation is involved in the detrimental effects of DOX-induced cardiotoxicity is not revealed. However, several studies have suggested potential pathways involved in the pro-death stimulus [27,39]. ERKs are known to play a role in cell survival and proliferation, but their dysregulation can contribute to cell death under certain conditions. The study by Chen et al. suggested that the activation of ERKs contributed to the detrimental effects of DOX on the myocardium [40]. These studies suggest that the activation of ERKs may be a key signaling mechanism transducing the pro-death stimulus of DOX-induced cardiotoxicity. However, it is important to note that the exact molecular events and downstream targets of ERKs in this context require further investigation for a comprehensive understanding of the signaling pathway. In summary, this study emphasizes the importance of further research to explore the potential of GSH in protecting against DOX-induced cardiotoxicity. There are several promising areas for future investigation. Firstly, exploring additional cardiac function biomarker genes and utilizing protein analysis techniques can provide deeper insights into the underlying mechanisms involved. Secondly, assessing markers of oxidative damage and mitochondrial function can help elucidate the protective effects of GSH. Additionally, non-invasive imaging techniques and pharmacokinetic studies of GSH can enhance our understanding and optimize therapeutic interventions. By pursuing these research directions, we can advance our knowledge and develop innovative strategies to mitigate the harmful effects of DOX on the heart.

Reviewer 2 Report
In this original article by Eun Ju Lee et al, the authors showed the protective role of Glutathione (GSH) from Doxorubicin (DOX) Induced cardiotoxicity in Human cardiac progenitor cells. Here are my concerns:
1. Fig 1A, x-axis of graph showed DOX concentration in nM (nano Molar) while line 59 of text showed the doses in µM (micro Molar). It is better to choose one of them and keep up the consistency.
2. Fig 1B, authors showed cell viability is more than 100 for GSH concentrations. Do they mean cell confluence? Because viability percentage of 100 means all the cells are alive.
3. Fig 1D and 1E, why the measuring has shown in two different magnification scale?
4. Fig 1F contains different font size for -/+ signs.
5. Fig 2A, why DOX concentration 250nM used rather than 500nM similar to the Fig1 is not clear.
6. Fig2B showed only 10% decrease in live cell % while 2C showed two times increase in apoptotic cells for DOX 250nM. The meaning and connection of these two bar charts are not clear.
7. Fig 2E missing the title of “Y-axis”. In line, 96 authors claim a figure named 2F that is completely missing in this Figure 2.
8. In Fig3B, again DOX 500nM used rather than 250nM and even lower for 4 B, D, F and 4H. The rational of using different doses need explanation. In Fig 4C lower panel that means 6H, not only the migration but also the cell density has hugely compromised in DOX 100nM.
9. In Fig5C, the graph needs real mg values of BW and if possible a HW/BW (heart weight to Body weight ratio) bar graph for better understanding of hypertrophy.
10. Fig 5D, for staining images, hearts from CON and DOX+GSH groups showed the level of cross section of hearts done at papillary muscles level. While hearts of DOX or GSH groups lack that indicating different level of cross section for hearts.
11. In method section, line 268, authors did not mention using any phosphatase inhibitors including sodium orthovanadate (Na3VO4). Without which the p-AKT and p-Erk blots are non-conclusive. Also another technical thing is if the PVDF has been activated with methanol before use as the quality of those blots are not as similar as other researchers of same field.
12. A western blot for Cytochrome C in Fig 2 and MnSOD in Fig 3 would be beneficial for the manuscript.
13. Overall, there are many places to improve including the mechanistic background showed in scheme of Fig6. How and why those pathways activated by ROS could be ventured. E.g. ROS is always presented with mitochondria, its energy coupling or its Ca2+ overload or even altering its dynamics. The English also needed to be checked.
e.g line 68. Cell cycle confirmed to be western
Author Response
- Comment and Suggestion for Authors: Fig 1A, x-axis of graph showed DOX concentration in nM (nano Molar) while line 59 of text showed the doses in µM (micro Molar). It is better to choose one of them and keep up the consistency.
Answer: Thank you for bringing this inconsistency to our attention. After careful consideration, we have decided to use nM (nano Molar) as the unit of measurement for DOX concentration in both the figure and the corresponding text. This will be reflected in the revised version of Fig 1A and any relevant mentions in the manuscript.
Revised text Line (60 - 61) : To confirm whether hCPCs were damaged by DOX, they were exposed to 100, 200, 500, 1000, 2000, 4000 and 5000 nM DOX for 24 h.
- Comment and Suggestion for Authors: Fig 1B, authors showed cell viability is more than 100 for GSH concentrations. Do they mean cell confluence? Because viability percentage of 100 means all the cells are alive.
Answer : We appreciate the reviewer's concern and the opportunity to further clarify our findings. While the observed increase in CCK8 assay results following GSH treatment suggests an elevation in mitochondrial NADH levels, resulting in a color change, we acknowledge the need to consider additional factors when interpreting this increase solely as an indicator of enhanced cell viability. Upon thorough analysis of the data, we found that although the CCK8 assay results exhibited an increase, there was no statistically significant rise in overall confluency and cell counts in the presence of GSH. To address this limitation, we have taken the reviewer's suggestion into account and have included Supplement figure 1. B, which showcases a representative photograph of the cells alongside the corresponding cell number graph. We want to emphasize that while the CCK8 assay indicated an increase, it is crucial to consider the overall cellular response, including cell confluency and counts. Based on your feedback, we have made the necessary modifications to our study.
Revised text Line (75 – 84) : During the process of establishing the optimal concentration of GSH treatment, the treatment with GSH led to an increase in cell viability (Figure 1B). To ascertain whether the enhanced viability in the presence of DOX-GSH is solely due to the protective effect of GSH or a standalone effect, an experiment was conducted following treatment with GSH alone. The results revealed that GSH treated did not affect significant alterations in cell confluency or cell number (Figure S1B, C), and there were no changes observed in cell cycle-related markers or major signal transduction (Figure S1D). These findings indicate that the addition of GSH in a normal cellular environment does not enhance the cell's proliferative capacity or functionality significantly. However, requires co-processing of GSH in a specific environment such as DOX.
- Comment and Suggestion for Authors: Fig 1D and 1E, why the measuring has shown in two different magnification scale?
Answer : We appreciate the reviewer's feedback. Upon careful examination of our data, we indeed found that Figure 1D displays the overall confluency of cells following treatment with doxorubicin and glutathione. This provides valuable information regarding the impact of these treatments on cell growth and proliferation. Furthermore, Figure 1E confirms that there are no significant morphological differences observed between the treatment groups, indicating that the presence of glutathione does not induce noticeable changes in cell appearance. Added text content to avoid misleading readers. We apologize for any confusion in our previous explanation and thank the reviewer for highlighting these important aspects of our experimental results.
Revised text Line (69 – 70) : It also improved the cell confluency (Figure 1D) and it did not lead to morphological differences (Figure 1E).
- Comment and Suggestion for Authors: Fig 1F contains different font size for -/+ signs.
Answer : We apologize for any confusion caused by this discrepancy and appreciate your meticulousness in reviewing the figure. We modified the figure based on the advice.
Revised Figure. 1F :
- Comment and Suggestion for Authors: Fig 2A, why DOX concentration 250nM used rather than 500nM similar to the Fig1 is not clear.
Answer: We apologize for any confusion caused by this inconsistency and appreciate the opportunity to provide clarification. In this experiment, cells were treated with 500 nM DOX, but the cell viability was too low to proceed with FACS or other experiments attaching antibodies. So the Dox concentration was reduced by half. However, we understand that the rationale behind this specific concentration difference was not adequately explained in the manuscript. To address this, we revised the manuscript to provide a clear and concise explanation for the use of 250nM DOX concentration in Fig 2A, while maintaining consistency with the dose range tested in the study.
Revised text Line (94 – 97) : In this experiment, the initial DOX concentration of 500 nM, as determined in Figure 1, resulted in lower cell viability, making it difficult to conduct further experiments. Experiments were conducted with 250 nM to confirm the apoptosis and antioxidant of cells caused by DOX.
- Comment and Suggestion for Authors: Fig2B showed only 10% decrease in live cell % while 2C showed two times increase in apoptotic cells for DOX 250nM. The meaning and connection of these two bar charts are not clear.
Answer : We apologize for any confusion caused by the lack of clarity in explaining the relationship between these two bar charts, and we appreciate the opportunity to address this issue. In Figure A, the percentages of live cells and apoptotic cells demonstrate a similar ratio of decrease and increase, indicating a correlation between the two. However, in Figures B and C, the quantitative graphs appear to have been affected by different y-axis ranges, potentially causing confusion in the interpretation of the data. We appreciate the reviewer's suggestion and have taken it into account by modifying the figures accordingly. We would like to express our gratitude for the valuable feedback provided.
Revision figure 2. B :
- Comment and Suggestion for Authors: Fig 2E missing the title of “Y-axis”. In line, 96 authors claim a figure named 2F that is completely missing in this Figure 2.
Answer: We apologize for these oversights and appreciate the opportunity to correct them. We advise according to the Fig. 2E modified.
Revised figure 2. E :
- Comment and Suggestion for Authors: In Fig3B, again DOX 500nM used rather than 250nM and even lower for 4 B, D, F and 4H. The rational of using different doses need explanation. In Fig 4C lower panel that means 6H, not only the migration but also the cell density has hugely compromised in DOX 100nM.
Answer: We appreciate your astute observations and the opportunity. In Fig. 3, we aimed to investigate the signaling pathways associated with the side effects of doxorubicin (DOX) and the protective effects of glutathione (GSH) at the concentrations selected in Fig. 1. However, in Fig. 4, we conducted an experiment to assess cell functions, including migration and tube formation ability. The wound healing assay, as you may be aware, evaluates the migratory capacity of cells by measuring their ability to close a wound created artificially. To eliminate the possibility of cell toxicity and loss of function due to decreased cell numbers, we treated cells with lower concentrations that did not exhibit cytotoxic effects. Furthermore, in Figure 4C, the density observed is contrary to Figures 1B and 1C, where lower density indicates reduced movement ability. Thank you for your valuable feedback.
Revised text Line (135 – 137): To evaluate the function of hCPCs after exposure to DOX, we selected a 100 nM concentration of DOX that did not affect the viability and exposed the cells treated with 100 nM DOX to GSH to conduct a wound-healing assay
- In Fig5C, the graph needs real mg values of BW and if possible a HW/BW (heart weight to Body weight ratio) bar graph for better understanding of hypertrophy.
Answer: Thank you for bringing this to our attention. In response to your feedback, we've added a Fig. 5D representation of the graph to include the actual value of body weight (BW). However, we regret to inform you that the heart-to-body weight ratio graphs could not be included in the study due to the unavailability of heart weight measurements at the time of the experiment. We apologize for any inconvenience caused and understand the potential significance of this data. If it is deemed necessary, we can explore the possibility of obtaining the heart weight measurements and adding the corresponding graphs in future experiments.
Revised figure 5. C :
- Fig 5D, for staining images, hearts from CON and DOX+GSH groups showed the level of cross section of hearts done at papillary muscles level. While hearts of DOX or GSH groups lack that indicating different level of cross section for hearts.
Answer : Thanks for the reviewer's keen observation. Based on this feedback, the figure was modified to include images of similar cross-sectional regions of the heart.
Revised Figure 5. D :
- In method section, line 268, authors did not mention using any phosphatase inhibitors including sodium orthovanadate (Na3VO4). Without which the p-AKT and p-Erk blots are non-conclusive. Also, another technical thing is if the PVDF has been activated with methanol before use as the quality of those blots are not as similar as other researchers of same field.
Answer: We thank you for pointing this out. We apologize for the mistake in labeling the materials and method. We have corrected this error in the revised manuscript. Thanks for the precise point, p-ERK and p-Akt were not observed without using the Protease and Phosphatase Inhibitor Cocktail.
Revised text Line (310 – 313) : After culturing in each group media, cells were lysed using RIPA lysis buffer (Thermo Fisher Scientific, Waltham, MA, USA) supplemented with either a protease inhibitor cocktail or protease and phosphatase inhibitor cocktail(Thermo Fisher Scientific, Waltham, MA, USA)
- Comment and Suggestion for Authors: A western blot for Cytochrome C in Fig 2 and MnSOD in Fig 3 would be beneficial for the manuscript.
Answer: We appreciate the reviewer's suggestion regarding conducting western blot experiments for Cytochrome C in Figure 2 and MnSOD in Figure 3, as it would be beneficial for the manuscript. However, due to the limited timeframe, we were unable to perform these experiments. If additional resources or time are made available, we will consider conducting these experiments in the future and include the results in the paper. These experiments could provide valuable insights and complement our current findings, making them a valuable suggestion for future research directions.
- Comment and Suggestion for Authors: Overall, there are many places to improve including the mechanistic background showed in scheme of Fig6. How and why those pathways activated by ROS could be ventured. E.g. ROS is always presented with mitochondria, its energy coupling or its Ca2+ overload or even altering its dynamics. The English also needed to be checked.
Answer: We appreciate the reviewer's concern and the opportunity to further clarify our findings. 1) Among the mechanisms implicated in cardiotoxicity, reactive oxygen species (ROS) contribute significantly. This experiment was conducted to address this issue, and modifications were made to the conclusion section to enhance the authors' comprehension.
2) We have enlisted the assistance of a native English speaker for English proofreading. If there are still numerous areas that require further editing, we are open to utilizing the Mdpi editing services to enhance the quality of the paper prior to publication.
Revised text Line (179 – 182) : DOX, an anticancer agent, exerts its cardiotoxic effects through multiple mechanisms involving not only ROS, but also calcium dysregulation and mitochondrial dysfunction. While these factors collectively contribute to cardiotoxicity, ROS can be considered a prominent factor in this process.

Reviewer 3 Report
In their MS entitled : “The Protective Role of Glutathione on Doxorubicin-Induced Cardiotoxicity in Human Cardiac Progeniter progenitor Cells”, Eun Ji Lee et al. underline the important salutary role of GSH in the recovery of Human Cardiac Progeniter progenitor Cells exposed to doxorubicin.
A major concern regarding this particular study is the fact that the progenitor cells used are routinely cultured in a medium already containing GSH. As the authors note in the methods: “ hCPCs were cultured in Ham’s F-12 medium containing 10 % heat- inactivated fetal bovine serum (FBS; Gibco, Thermo Fisher Scientific, Carlsbad, CA, USA), 1 % penicillin/streptomycin (P/S, Welgene, Daegu, Republic of Korea), 0.2 mM of L-glutathione (Sigma-Aldrich #G4251, St. Louis, CA, USA), 20 ng/mL of recombinant human basic fibroblast growth factor (rb-FGF; PeproTech, Rocky Hill, NJ, USA), and 0.005 unit/mL of human erythropoietin (hEPO; R&D Systems, Minneapolis, MN, USA).
The authors do not report any serum deprivation period before performing their experiments-thus when all treatments are performed the medium is the one described above? The deprivation period is essential so that all responses observed can be attributed exclusively to the stimuli under investigation and not to the serum or in this case and so many other agents present in the medium!
The fact that GSH is already present and obviously essential for the culture of these cells raises significant concerns.
Secondly, in Figure 1A in the presence of increasing concentrations of GSH the viability of the cells was increased. If this is the case, then the increase of viability in the presence of DOX-GSH could be attributed to this effect rather than to a salutary effect of GSH.
In Figs 1D and 1E there is no image for the state of the cells in the presence solely of GSH-this should be included! There is also no bar indicating magnification.
In Fig 1G there is also no sample of GSH-treated cells so that the effect of GSH alone can be estimated and detected on cyclin E and CDK4 expression levels.
In Fig 2A the effect of GSH alone on apoptosis is also missing ( FACS using annexin V/PI staining kit).
In Fig 3A the effect of GSH alone on ERKs and AKT phosphorylation levels is also missing. This should also be included since it may affect the observed responses. Authors should also note against which phosphorylated residue of Akt is the antibody they used in the image depicted in fig. 3A: S473 or T308?
Densitometric analysis of all western blots should be provided.
In Fig.3B values expressed are the mean ± standard derivation but no P<0.05 as compared to the 111 DOX group> value is illustrated. Does this mean that no statistically significant effect was observed in the presence of U0126?
ERKs as well as Akt are established as kinases exerting a beneficial-salutary effect under stressful conditions in a number of experimental settings. Since authors have correlated ERKs phosphorylation with the detrimental effect of DOX what is the signaling mechanism transducing this pro-death stimulus?
Regarding the cardiomyopathy model used, how many C57BL/6 mice were intraperitoneally injected with either doxorubicin (20 mg/kg) or glutathione (100 mg/kg) ? This important information is also missing.
The English language also needs extensive editing throughout the text.
Moderate editing of the English language is needed.
Author Response
- Comment and Suggestion for Authors: A major concern regarding this particular study is the fact that the progenitor cells used are routinely cultured in a medium already containing GSH. As the authors note in the methods: “ hCPCs were cultured in Ham’s F-12 medium containing 10 % heat- inactivated fetal bovine serum (FBS; Gibco, Thermo Fisher Scientific, Carlsbad, CA, USA), 1 % penicillin/streptomycin (P/S, Welgene, Daegu, Republic of Korea), 0.2 mM of L-glutathione (Sigma-Aldrich #G4251, St. Louis, CA, USA), 20 ng/mL of recombinant human basic fibroblast growth factor (rb-FGF; PeproTech, Rocky Hill, NJ, USA), and 0.005 unit/mL of human erythropoietin (hEPO; R&D Systems, Minneapolis, MN, USA).
The authors do not report any serum deprivation period before performing their experiments-thus when all treatments are performed the medium is the one described above? The deprivation period is essential so that all responses observed can be attributed exclusively to the stimuli under investigation and not to the serum or in this case and so many other agents present in the medium!
The fact that GSH is already present and obviously essential for the culture of these cells raises significant concerns.
Answer : Thank you for your valuable question. First, in this study, serum deficiency was not separately performed. 1) The primary objective of this study was to investigate the mechanisms by which cells defend against cardiotoxicity induced by doxorubicin in a normal environment. Therefore, the focus was on enhancing the viability and functionality of cells that were compromised by doxorubicin under basic culture conditions. 2) We have also been concerned about your opinion. During the process of setting the concentration of doxorubicin and GSH, we tried to proceed with serum-free conditions. As a result, it did not lead to significant differences. We have interpreted that a certain concentration of GSH is essential for cell culture, and we believe that additional GSH is required to improve the viability and recovery of cells reduced by doxorubicin.
Revised supplement figure 1. A :
- Comment and Suggestion for Authors: Secondly, in Figure 1A in the presence of increasing concentrations of GSH the viability of the cells was increased. If this is the case, then the increase of viability in the presence of DOX-GSH could be attributed to this effect rather than to a salutary effect of GSH.
Answer : We appreciate the reviewer's concern and the opportunity to further clarify our findings. In the case of the group treated with GSH alone, an additional experiment was performed to assess the potential toxicity of the treatment on cells. The observed increase in CCK8 assay results following GSH treatment does indeed indicate an elevation in mitochondrial NADH levels, resulting in a color change from colorless to colored. However, we acknowledge that the interpretation of this increase solely as an indicator of enhanced cell viability might be misleading without considering other factors.
In light of the reviewer's feedback, we have reevaluated our results and recognize the importance of assessing overall confluency and cell counts in addition to the CCK8 assay. Upon reviewing the data, we found that although the CCK8 assay results showed an increase, there was no statistically significant increase in overall confluency and cell counts in the presence of GSH.
To address this limitation, we have submitted Figure 1D – F, which includes a representative photograph of the cells along with the corresponding cell number graph. This data provides a comprehensive view of the cellular response to GSH treatment, allowing for a more accurate interpretation of the experimental outcomes (Supplement data 1. B - C). We want to emphasize that while the CCK8 assay showed an increase, it is essential to consider the overall cellular response, including cell confluency and counts. Based on other results and the premise that GSH exhibits reactivity with toxins, we proceeded with the experiment to explore its potential protective effects.
Revised Supplement figure 1. B – C :
Revised text Line (76 – 80) : To ascertain whether the enhanced viability in the presence of DOX-GSH is solely due to the protective effect of GSH or a standalone effect, an experiment was conducted following treatment with GSH alone. The results revealed that GSH treated did not affect significant alterations in cell confluency or cell number (Figure S1B, C).
- Comment and Suggestion for Authors: In Figs 1D and 1E there is no image for the state of the cells in the presence solely of GSH-this should be included! There is also no bar indicating magnification.
Answer : We appreciate the reviewer's keen observation. In response to this feedback, we have revised the figures to include the image depicting the state of the cells in the presence solely of GSH. The new version of Figs 1D and 1E now includes the additional image, providing a comprehensive representation of the experimental conditions.
Revised figure 1. D – F :
- Comment and Suggestion for Authors: In Fig 1G there is also no sample of GSH-treated cells so that the effect of GSH alone can be estimated and detected on cyclin E and CDK4 expression levels.
Answer : Thank you for your insightful comment regarding Fig 1G. To address this concern, we conducted a comparative analysis between a control group and cell samples treated exclusively with GSH. In response to the reviewer's feedback, we have included supplementary figures in our revised manuscript to provide visual representations of the experimental results. Additionally, we have incorporated the corresponding findings into the content of the results section.
Revised supplement figure 1. D :
Revised text Line (79 – 81) : The results revealed that GSH treated did not affect significant alterations in cell confluency or cell number (Figure S1B, C), and there were no changes observed in cell cycle related markers or major signal transduction (Figure S1D).
- Comment and Suggestion for Authors: In Fig 2A the effect of GSH alone on apoptosis is also missing (FACS using annexin V/PI staining kit).
Answer : We sincerely appreciate your insightful feedback, as it helps us enhance the quality and comprehensiveness of our research. To address this concern, we added supplement figure to include the assessment of apoptosis induced by GSH alone. Furthermore, we have integrated the relevant outcomes into the results section to ensure that the corresponding findings are appropriately represented.
Revised supplement figure 2. A – B :
Revised text Line (100 – 102) : We performed an experiment comparing the GSH alone treated group to the control, also. As a result, no significant difference between the two groups could be confirmed (Figure S2. A, B).
- Comment and Suggestion for Authors: In Fig 3A the effect of GSH alone on ERKs and AKT phosphorylation levels is also missing. This should also be included since it may affect the observed responses. Authors should also note against which phosphorylated residue of Akt is the antibody they used in the image depicted in fig. 3A: S473 or T308? – 실험
Answer : We sincerely appreciate your meticulous review and constructive feedback, as it helps us improve the quality and clarity of our research. To improve our analysis, we incorporated the use of the pAKT S473 antibody in our study. Consequently, we made modifications to the Materials & Methods section to reflect this change. Furthermore, we included the experimental results obtained from the GSH group to provide a more comprehensive understanding of the impact of glutathione (GSH) in our study. These updates ensure that our findings are more robust and contribute to the overall integrity of the research.
Revised Supplement figure 1. D :
Revised text Line (125 – 127) : Similar to the previous findings, the cell group treated with GSH did not exhibit a significant difference compared to the control group (Figure S1. D).
- Comment and Suggestion for Authors: Densitometric analysis of all western blots should be provided.
Answer : We appreciate your valuable input, as it helps us improve the scientific rigor and clarity of our research. In response to your feedback, we ensured that densitometric analysis is performed and included for all Western blots presented in the manuscript.
Revised Figure 1. G and Figure 3. A :
- Comment and Suggestion for Authors: In Fig.3B values expressed are the mean ± standard derivation but no P<0.05 as compared to the 111 DOX group> value is illustrated. Does this mean that no statistically significant effect was observed in the presence of U0126?
Answer : We apologize for the oversight in not including the indication of statistical significance in Fig. 3B. The meaningful mark has been removed from the file delivery process. It was present in the submitted PPT and PDF files, but it was not indicated in the paper, which has been corrected.
Revised figure 3. B :
- Comment and Suggestion for Authors: ERKs as well as Akt are established as kinases exerting a beneficial-salutary effect under stressful conditions in a number of experimental settings. Since authors have correlated ERKs phosphorylation with the detrimental effect of DOX what is the signaling mechanism transducing this pro-death stimulus?
Answer : First of all, we appreciate the reviewer’s insightful comment. In our study, while the activation of ERK signaling was associated with the detrimental effect of DOX-induced cardiotoxicity, we acknowledge the need for a deeper understanding of the specific signaling pathways involved in mediating this pro-death stimulus. Although the signaling mechanisms by which ERKs phosphorylation is involved in the detrimental effects of DOX-induced cardiotoxicity have not been fully elucidated, several studies have proposed potential pathways related to death-promoting stimuli. However, the precise molecular events and downstream targets of ERK need further investigation in this context. Approving your opinion, I added the content and reference to the discussion part.
Revised text Line (211 – 222) : Although we found that the protective effects of GSH in vitro are related to the pERK pathway to cardiotoxicity, the signaling mechanism by which ERKs phosphorylation is involved in the detrimental effects of DOX-induced cardiotoxicity is not revealed. However, several studies have suggested potential pathways involved in the pro-death stimulus [27,39]. ERKs are known to play a role in cell survival and proliferation, but their dysregulation can contribute to cell death under certain conditions. The study by Chen et al. suggested that the activation of ERKs contributed to the detrimental effects of DOX on the myocardium [40]. These studies suggest that the activation of ERKs may be a key signaling mechanism transducing the pro-death stimulus of DOX-induced cardiotoxicity. However, it is important to note that the exact molecular events and downstream targets of ERKs in this context require further investigation for a comprehensive understanding of the signaling pathway.
- Comment and Suggestion for Authors: Regarding the cardiomyopathy model used, how many C57BL/6 mice were intraperitoneally injected with either doxorubicin (20 mg/kg) or glutathione (100 mg/kg)? This important information is also missing.
Answer : Thank you for drawing our attention to this oversight, and we sincerely apologize for any confusion it may have caused. We have added the omitted content.
Revised text Line (335 – 338) : For the doxorubicin-induced cardiomyopathy model, C57BL/6 mice were randomized into three group and intraperitoneally injected with 1 x PBS (n = 10), doxorubicin (n = 10, 20 mg/kg) or glutathione (n = 10, 100 mg/kg) based on reported studies[17,42,43].
- Comment and Suggestion for Authors: The English language also needs extensive editing throughout the text.
Answer : Thank you for your comment. We have made revisions to improve the overall content of the manuscript. Additionally, we have enlisted the assistance of a native English speaker for English proofreading. If there are still numerous areas that require further editing, we are open to utilizing the Mdpi editing services to enhance the quality of the paper prior to publication.

Round 2
Reviewer 1 Report
The Authors have extensively revised the manuscript and have satisfactorily addressed my concerns. However, though in Section 4.12 they have specified the genes analyzed and the sequences of the primers used, they did not provide the cycling conditions/program(s) used for gene amplification.
English language still requires some editing (e.g., "a concentration that does not death the cells" should be replaced by "a concentration that does not kill the cells" or "a concentration that does not induce cell death", among other details).
Author Response
- The Authors have extensively revised the manuscript and have satisfactorily addressed my concerns. However, though in Section 4.12 they have specified the genes analyzed and the sequences of the primers used, they did not provide the cycling conditions/program(s) used for gene amplification.
Answer : Thank you for your valuable response. We are particularly grateful that you found our revisions satisfactory. Also, we apologize for the oversight and appreciate you bringing this to our attention. In response to your comment, we have revised the manuscript accordingly to include the necessary details of the cycling conditions/program(s) employed for gene amplification in Section 4.12.
Revision text Line (364 – 366) : The qRT-PCR protocol was begun with 95 °C incubation for 10 min. There followed by 45 cycles of 95 °C for 10 s, 60 °C for 10 s and 72 °C for 10 s. Then, followed by melting at 95 °C for 10 s, 60 °C for 60 s.
- English language still requires some editing (e.g., "a concentration that does not death the cells" should be replaced by "a concentration that does not kill the cells" or "a concentration that does not induce cell death", among other details).
Answer : We sincerely appreciate your feedback, as it helps us enhance the quality and readability of our work. We have taken your comments into consideration and address the English language issues accordingly. We have also revised our manuscript and edited it for language and grammar with the help of a native English speaker. The parts that have minor changes according to grammar and notation are marked in green.

Reviewer 2 Report
The comments of the authors are satisfactory.
There is no need to put numbers under the blots. If needed a molecular weight indication of kDa could be used for better view.
Author Response
- There is no need to put numbers under the blots. If needed a molecular weight indication of kDa could be used for better view.
Answer : We appreciate your time and effort in reviewing our work and providing insightful comments. We are pleased to submit the revised version of our manuscript, incorporating the improvements suggested by you and the other reviewers.
In response to another reviewer's feedback during the first revision, densitometric analysis of all western blots has been performed and the results have been included as supplementary material.
However, in response to your suggestion, we agree that the inclusion of numerical values directly beneath the blots may not be necessary for a better visual presentation.
Therefore, we will remove a clear description of numerical values directly beneath the blots, as suggested, and supplement data will include densitometric analysis.
Once again, we would like to express our gratitude for your expertise and constructive feedback. We look forward to any further suggestions or comments you may have on the revised manuscript.

Reviewer 3 Report
Authors have performed extensive revisions. However, the major principal concern on the lack of a sufficient serum deprivation period before conducting the experiments still stands. This is imperative when signaling mechanisms are under investigation and therefore the justification given : ". Therefore, the focus was on enhancing the viability and functionality of cells that were compromised by doxorubicin under basic culture conditions" is not "valid". One cannot overlook the existence of serum in the medium...
Extensive editing has been performed.
Author Response
Reviewer’s Responses to Questions.
- Authors have performed extensive revisions. However, the major principal concern on the lack of a sufficient serum deprivation period before conducting the experiments still stands. This is imperative when signaling mechanisms are under investigation and therefore the justification given : "Therefore, the focus was on enhancing the viability and functionality of cells that were compromised by doxorubicin under basic culture conditions" is not "valid". One cannot overlook the existence of serum in the medium.
Answer : We appreciate the reviewer's concern regarding the lack of a sufficient serum deprivation period before conducting the experiments. We acknowledge the crucial importance of considering serum conditions when studying signaling mechanisms, and we completely agree with the reviewer that potential impacts of serum components on our results need careful consideration.
However, when conducting all experiments using cardiac progenitor cells, they almost exhibit low survival rates under serum deprivation conditions. In our research, we utilized cardiovascular progenitor cells, and cell viability played a crucial role in performing experiments such as tube formation and migration assays. To address this issue and ensure the reliability of our experimental results, we referred to relevant literature for additional insights. A considerable body of literature focused on cardiac progenitor cells typically conducts their experiments in environments containing serum. This practice is widespread and well-founded due to the recognition of serum's importance in promoting cell viability and supporting cellular processes [26,27]. By considering the specific characteristics of cardiovascular progenitor cells, acknowledging the difficulties in serum deprivation, and aligning our approach with the prevailing practices in similar research, we aimed to ensure the reliability and meaningful interpretation of our experimental data.
Additionally, we conducted experiments related to doxorubicin in cardiomyocytes, including H9C2 and AC16, and observed increase in pERK levels [28,29].
However, we have taken the reviewer's comments to heart and recognized the importance of conducting experiments in a serum deprivation environment. As a result, we proceeded with the experiment as depicted in Supplement Figure 1a. Despite reviewer's concerns, the results revealed no significant difference between the conditions with or without serum.
Incorporating this crucial information, we have made every effort to prevent confusion among our readers and strengthen the credibility of our research. We firmly believe that our comprehensive clarifications, along with the inclusion of additional references, will effectively address any doubts raised by the reviewer.
We extend our deepest gratitude to the reviewer for providing insightful feedback, which has significantly contributed to refining and enhancing the scientific merit of our manuscript. Your input has been invaluable, and we are committed to making the necessary improvements to produce a robust and reliable study.
Revised text Line (69 – 82) : Numerous studies have explored the effects of different conditions by conducting experiments in serum-free environments to understand their potential impact on cell behavior. In line with this, we performed cell viability assessments in a serum deprivation setting to verify the significance of serum conditions. Remarkably, our findings exhibited a similar pattern of results in both settings, suggesting that serum components did not significantly influence the drug's efficacy (Figure S1A). It is important to note that attempting to induce serum deprivation in cardiac progenitor cells led to an exceptionally low survival rate, making it challenging to carry out further experiments, including functional evaluations. In order to address these limitations and ensure the feasibility of our investigations, we extensively reviewed relevant literature [26,27]. The majority of these references had con-ducted their experiments in environments containing serum. Considering the well-established difficulties with serum deprivation and the wealth of supportive evidence from the literature, we proceeded with conducting all experiments in environments containing serum.
- De Angelis, A.; Piegari, E.; Cappetta, D.; Marino, L.; Filippelli, A.; Berrino, L.; Ferreira-Martins, J.; Zheng, H.; Hosoda, T.; Rota, M.; et al. Anthracycline cardiomyopathy is mediated by depletion of the cardiac stem cell pool and is rescued by restoration of progenitor cell function. Circulation 2010, 121, 276-292, doi:10.1161/CIRCULATIONAHA.109.895771.
- Sebastiao, M.J.; Serra, M.; Pereira, R.; Palacios, I.; Gomes-Alves, P.; Alves, P.M. Human cardiac progenitor cell activation and regeneration mechanisms: exploring a novel myocardial ischemia/reperfusion in vitro model. Stem Cell. Res. Ther. 2019, 10, 77, doi:10.1186/s13287-019-1174-4.
- Park, E.J.; Kwon, H.K.; Choi, Y.M.; Shin, H.J.; Choi, S. Doxorubicin induces cytotoxicity through upregulation of pERK-dependent ATF3. PLoS One 2012, 7, e44990, doi:10.1371/journal.pone.0044990.
- Liu, J.; Mao, W.; Ding, B.; Liang, C.S. ERKs/p53 signal transduction pathway is involved in doxorubicin-induced apoptosis in H9c2 cells and cardiomyocytes. Am J Physiol Heart Circ Physiol 2008, 295, H1956-1965, doi:10.1152/ajpheart.00407.2008.

Round 3
Reviewer 3 Report
After authors inclusion of comments regarding serum deprivation period necessity and the particular characteristics of progenitor cells, the MS has improved significantly.
No issues detected.